# Germline BRCA Mutation and Clinical Outcomes in Breast Cancer Patients Focusing on Survival and Failure Patterns: A Long-Term Follow-Up Study of Koreans

**DOI:** 10.3390/medicina56100514

**Published:** 2020-10-01

**Authors:** Hakyoung Kim, Doo Ho Choi, Won Park

**Affiliations:** 1Department of Radiation Oncology, Korea University Guro Hospital, Korea University College of Medicine, Seoul 02841, Korea; khk614@gmail.com; 2Department of Radiation Oncology, Samsung Medical Center, Sungkyunkwan University School of Medicine, Seoul 06351, Korea; wonro.park@samsung.com

**Keywords:** breast cancer, BRCA mutation, survival analysis, recurrence

## Abstract

*Background and Objectives*: This study aimed to evaluate the effect of a BRCA mutation on survival and failure patterns, focusing on the risk of ipsilateral recurrence and contralateral breast cancer in patients. *Materials and Methods*: We retrospectively reviewed medical records of 300 patients with breast cancer who underwent genetic screening for *BRCA1/2* genes and were treated at Samsung Medical Center between 1 January 2000 and 31 December 2010. Ultimately, clinical outcomes of 273 patients were analyzed. *Results*: The median follow-up duration was 102 months (range, 1 to 220 months). Patients with *BRCA1/2*-mutated tumors had a shorter 10-year disease-free survival (DFS) rate compared to those with non-mutated tumors (62.8% vs. 80.0%, *p* = 0.02). Regarding failure patterns, patients with *BRCA1/2*-mutated tumors showed a higher incidence of contralateral breast cancer than those with non-mutated tumors (*BRCA1/2* non-mutated vs. mutated tumors: 4.9% vs. 26.0%, *p* < 0.001). BRCA mutation status remained a significant prognostic factor for contralateral breast recurrence-free survival (HR: 4.155; 95% CI: 1.789–9.652; *p* = 0.001). Korean patients with a BRCA mutation showed inferior DFS compared to those without a BRCA mutation. *Conclusions*: BRCA mutation status is a strong predictor of recurrence in contralateral breast cancer. Strategies such as prophylactic treatment and active surveillance should be discussed with breast cancer patients who have a BRCA mutation.

## 1. Introduction

A BRCA mutation is a mutation in either the *BRCA1* or *BRCA2* gene, both of which are tumor-suppressor genes. A harmful mutation in these genes might provoke a hereditary breast–ovarian cancer syndrome in affected persons. *BRCA1* and *BRCA2* mutations have been found in 5–10% of all breast cancers and in up to 20–25% of tumors in patients with a family of breast and/or ovarian cancer [1]. However, the impact of these gene mutations on women might be more profound.

According to previous studies on BRCA mutations, *BRCA1* mutation carriers are more likely to be triple negative with a lower estrogen receptor level, higher histological grade, and higher proliferation index than patients who have no such mutation. On the other hand, *BRCA2* mutation carriers are more likely to be estrogen-receptor positive, similar to those with sporadic tumors [2,3,4,5]. Whether a BRCA mutation in breast cancer is associated with poor prognosis remains controversial. Some studies have demonstrated that *BRCA1/2* mutation carriers have a worse survival outcome [6], while others have shown that *BRCA1/2* mutation carriers have similar or better survival than non-carriers [7,8,9]. In meta-analysis, *BRCA1* mutation carriers show decreased overall survival (OS) and progression-free survival (PFS) while *BRCA2* mutation carriers do not. In terms of failure patterns, several studies have suggested that the recurrence rate in *BRCA1*/*BRCA2* mutation carriers is not increased compared to that in non-carriers [10,11,12,13]. Other studies have compared ipsilateral and/or contralateral breast recurrence in *BRCA1* and *BRCA2* mutation carriers and patients with sporadic cancers. These studies have consistently found an elevated risk of contralateral breast cancer in BRCA mutation carriers [14,15,16]. However, whether the risk of ipsilateral recurrence is higher in women with a BRCA mutation remains controversial. Current treatment for BRCA mutation-associated breast cancer is not different from that for sporadic breast cancer.

The purpose of this study was to evaluate the effect of a BRCA mutation on survival and recurrence rate, focusing on risk of ipsilateral recurrence and contralateral breast cancer in breast cancer patients who underwent genetic screening for a *BRCA1/2* mutation and were treated at the Samsung Medical Center.

## 2. Materials and Methods

### 2.1. Patients

After obtaining approval from our Institutional Review Board (IRB), we retrospectively reviewed medical, pathology, and radiotherapy records of 300 patients with breast cancer who underwent genetic screening for *BRCA1/2* mutation and were treated at Samsung Medical Center between 1 January 2000 and 31 December 2010. Genetic screening was performed for those who met the criteria of National Health Insurance System of Korea, including breast cancer with family history, bilateral breast cancer, breast cancer with family history of ovarian cancer, male breast cancer, and diagnosed before 40 years old. Among these, patients with bilateral breast cancer (25 patients) and those with history of previous or concurrent malignancy except for thyroid cancer at the time of diagnosis (*n* = 2) were excluded from this study. Ultimately, clinical outcomes of 273 patients were analyzed, with emphasis on the recurrence rate, including risks of ipsilateral recurrence and contralateral breast cancer according to BRCA mutation status. All tumors were staged based on the seventh edition of the American Joint Committee on Cancer (AJCC) tumor staging criteria. Revised Response Evaluation Criteria In Solid Tumors (RECIST) guidelines (version 1.1) were used for tumor response evaluation.

### 2.2. Treatment Scheme

After genetic screening, 218 (79.9%), 42 (15.4%), and 13 (4.7%) patients underwent breast-conserving surgery (BCS), modified-radical mastectomy (MRM), and total mastectomy (TM), respectively. Sentinel lymph node biopsy (SLNB) and axillary lymph node dissection (ALND) were performed in 141 (51.6%) and 121 (43.3%) patients, respectively. The remaining 11 (5.1%) patients who were diagnosed with ductal carcinoma in situ underwent BCS only without lymph node sampling. In the current study, none of the patients underwent prophylactic contralateral mastectomy. The median number of sampled lymph nodes (LNs) was 8 (range, 1–49). Neoadjuvant chemotherapy was delivered to 17 (6.2%) patients while adjuvant chemotherapy was delivered to 190 (69.6%) patients except for three patients who were indicated but refused further treatment. The most common regimen was anthracycline plus cyclophosphamide, followed by taxane. All patients who were hormone receptor positive at initial clinical diagnosis (*n* = 193, 70.7%) received adjuvant hormone treatment. Among patients who were HER2/neu receptor positive at initial diagnosis (*n* = 37, 13.6%), 27 received adjuvant herceptin treatment. Adjuvant radiotherapy was delivered to 234 (85.7%) patients after surgery or at the end of chemotherapy. RT dose delivered to the whole breast or chest wall was most frequently at 50 Gy in 25 fractions. Later, the primary tumor bed was boosted most commonly with 9–15 Gy in 3–5 fractions according to the surgical resection margin status.

### 2.3. Histopathology and BRCA Mutation Evaluation

After the operation, hematoxylin and eosin-stained slides of sections of gross residual tumor and LN(s) were assessed for a total of 273 patients by an experienced pathologist. The following molecular subtypes were classified according to ER, PR, HER2, and Ki-67 receptor status: luminal A (ER+ and/or PR+, HER2−, and negative Ki-67), luminal B (ER+ and/or PR+ and HER2+) or (ER+ and/or PR+ and HER2–, and positive Ki-67)), HER2-enriched (ER−, PR−, and HER2+), and triple-negative (ER−, PR−, and HER2−) [17].

BRCA mutation analysis was conducted mainly at the Department of Laboratory Medicine and Genetics at Samsung Medical Center with the cooperation of three other DNA testing laboratories, all of which are certified annually by the Korean Institute of Genetic Testing Evaluation. Genomic DNA was extracted and purified from peripheral blood leukocytes. The whole exons and the flanking intrinsic sequences of the *BRCA1* gene or *BRCA2* gene were amplified by polymerase chain reaction. The amplified products were directly sequenced, and the sequences were then compared with reference sequences using Sequencher software (Gene Codes Co., Ann Arbor, MI, USA). The nomenclature for BIC (Breast Cancer Information Core) traditional mutations was used, based on U14680 (*BRCA1*) and U43746 (*BRCA2*). In addition, all mutations were described according to HUGO-approved systematic nomenclature (nomenclature for the description of sequence variations, Human Genome Variation Society. http://www.hgvs.org/mutnomen/). HUGO-approved mutation nomenclature of *BRCA1* (GenBank accession no. NP_009225.1) and *BRCA2* (GenBank accession no. NP_000050.2) defined the A of the ATG translation initiation codon as nucleotide +1. Splicing-defect mutations in intronic region were described at the genomic DNA level using GenBank genomic reference sequence NC_000017.10 (*BRCA1*) and NC_000013.10 (*BRCA2*). In addition, variants of unknown significance were excluded. Genetic testing of high-risk breast cancer patients was approved by the IRB of Samsung Medical Center (2010-09-006-001).

### 2.4. Statistical Analysis

Overall survival (OS) was defined as the time from the date of the surgery until the date of death from any cause or the latest documented follow-up. Disease-free survival (DFS) was defined as the time from the date of the surgery until the date of the first documented recurrence or the latest follow-up. To compare clinicopathologic characteristics according to BRCA mutation status, Chi-squared or Fisher’s exact tests was used. Survival rates were estimated using the Kaplan–Meier method and compared using log-rank tests. Factors that showed a probability value of <0.1 and those that were thought to be relevant were entered into a Cox proportional hazard regression analysis to determine independent prognostic factors. A *p* ≤ 0.05 was regarded as indicative of statistical significance in two-tailed tests. Statistical analysis was performed using SPSS software, standard version 24.0 (IBM Corporation, Armonk, NY, USA).

## 3. Results

### 3.1. Patient Characteristics

Characteristics of patients, tumors, and treatment are summarized in Table 1. The median age of the population was 43 years (range, 21 to 70 years). Of a total of 273 patients, 251 (91.9%) were classified as clinical Tis-2 and 22 (8.1%) were classified as clinical T3–4. Clinically, regional lymph node involvement was present in 101 (37.0%) patients. One hundred and ninety-three patients (70.7%) were hormone receptor positive and 37 patients (13.6%) were HER2 receptor positive. Regarding molecular subtypes, 124 (45.4%), 69 (25.3%), 18 (6.6%), and 62 (22.7%) patients had luminal A, luminal B, HER2-enriched, and triple-negative subtypes, respectively. Regarding pathology, a large proportion (218 patients, 79.9%) of patients had invasive ductal carcinoma.

### 3.2. Pathologic Characteristics According to BRCA Mutation Status

Pathologic characteristics according to BRCA mutation status are described in Table 1. *BRCA1/2*-mutated tumors and molecular subtypes showed significant correlations (*p* = 0.001). In addition, negative hormonal receptor (*p* = 0.004), negative HER2/neu receptor (*p* = 0.008), positive Ki-67 receptor status (*p* = 0.013), and higher histologic grade (*p* = 0.012) were associated with *BRCA1/2*-mutated tumors.

### 3.3. Survival Outcomes and Failure Patterns According to BRCA Mutation Status

The median follow-up duration was 102 months (range, 1 to 220 months). Survival outcomes according to BRCA mutation status are shown in Table 2. Ten-year OS rates did not show significant differences according to BRCA mutation status (non-mutated vs. *BRCA1/2*-mutated tumors: 96.2% vs. 98.0%, *p* = 0.844). In contrast, *BRCA1/2*-mutated tumors showed a significant difference in 10-year DFS rates (80.0% vs. 62.8%, *p* = 0.02, Figure 1). In further analysis, only the *BRCA1* mutation showed decreased DFS (non-mutated vs. *BRCA1*-mutated tumors: 78.6% vs. 62.1%, *p* = 0.031) while OS was similar between the two groups (*p* = 0.503). In contrast, the *BRCA2* mutation was not associated with decreased OS (*p* = 0.253) or DFS (*p* = 0.197).

Ipsilateral recurrence and contralateral breast cancer were found in 13 (4.8%) and 24 (8.8%) patients, respectively. Regional recurrence was found 11 (4.0%) patients while distant metastasis was noted in 18 (6.6%) patients. The two most common sites were the lung (*n* = 7) and bone (*n* = 4). According to BRCA mutation status, *BRCA1/2*-mutated tumors showed a higher incidence of contralateral breast recurrence compared to non-mutated tumor (non-mutated vs. *BRCA1/2*-mutated tumors: 4.9% vs. 26.0%, *p* < 0.001). Statistically-significant difference was found in 10-year contralateral breast recurrence-free survival (RFS) rates (non-detected vs. detected group: 95.0% vs. 73.7%, *p* < 0.001). However, no significant difference was found in the incidence of ipsilateral breast recurrence (4.5% vs. 6.0%, *p* = 0.649) or 10-year ipsilateral breast RFS rates (93.8% vs. 93.0%, *p* = 0.926).

### 3.4. Prognostic Factors for Disease-Free Survival and Breast Recurrence

Clinical N stage (*p* < 0.001), BRCA mutation status (*p* = 0.02), hormonal receptor status (*p* = 0.028), and Ki-67 status (*p* = 0.013) were significantly associated with DFS on univariate analysis (Table 3). On multivariate analysis, clinical N stage remained as a significant prognostic factor for DFS (HR: 2.078; 95% CI: 1.043–4.143; *p* = 0.038) (Table 4). In terms of breast recurrence, BRCA mutation status (*p* < 0.001), hormonal receptor status (*p* = 0.021), and histologic grade (*p* = 0.035) were significantly associated with contralateral breast RFS on univariate analysis. However, there was no significant prognostic factor for ipsilateral breast RFS (Table 3). On multivariate analysis, only BRCA mutation status remained as a significant prognostic factor for contralateral breast RFS (HR: 4.155; 95% CI: 1.789–9.652; *p* = 0.001) (Table 4).

## 4. Discussion

Patients with deleterious mutations in either *BRCA1* or *BRCA2* have about five times higher risk of breast cancer that those without such mutations. However, the effect of BRCA mutation on survival and the recurrence rate with emphasis on the risk of ipsilateral recurrence and contralateral breast cancer in breast cancer patients remains controversial. Through this single institutional study with an overall median follow-up of 8.5 years, we showed that *BRCA1/2*-mutated tumors were associated with negative hormonal and HER2/neu and positive Ki-67 receptor status, and higher histologic grade compared to non-mutated tumors. A significant correlation between BRCA mutation and the risk of secondary ovarian cancer was also found (non-mutated vs. *BRCA1/2*-mutated tumors: 2.2% vs. 14.0%, *p* = 0.001).

In the aspect of BRCA mutation status as a prognostic factor, many previous studies have shown conflicting results. Specifically, Robson et al. [6] have reported that 10-year breast-cancer-specific survival is significantly worse in *BRCA1* mutation carriers than that in non-carriers (62% vs. 86%, *p* < 0.001), but not in B*RCA2* mutation carriers (84% vs. 86%, *p* = 0.76). However, *BRCA1* mutation status was predictive of a worse outcome in those who did not receive chemotherapy. Similarly, Goodwin et al. [13] have proved that the survival of *BRCA1* carriers who are administered chemotherapy is similar to that of non-carriers, although the survival of *BRCA1* carriers is worse in the absence of chemotherapy (HR: 1.97; 95% CI: 0.65 to 5.94). In contrast, Rennert et al. [7] have shown no difference in 10-year survival rates among *BRCA1* mutation carriers, *BRCA2* mutation carriers, and non-carriers. Brekelmans et al. [8,9] have also reported that breast-cancer-specific survival is not different between *BRCA1-*mutation carriers and sporadic controls (HR: 1.29, 95% CI: 0.85–1.97). Later, Huzarski et al. [18] performed a study to estimate 10-year OS rates for patients with early-onset breast cancer with or without *BRCA1* mutation and identify prognostic factors among those with *BRCA1*-positive breast cancer. They concluded that the survival rate among women with *BRCA1* mutation was similar to that of patients without *BRCA1* mutation (*p* = 0.41). Among *BRCA1* mutation carriers, positive lymph node status was a strong predictor of mortality (adjusted HR: 4.1; 95% CI: 1.8 to 8.9; *p* < 0.001). A recent prospective cohort study [19] has shown that breast cancer patients with young-onset who carry a BRCA mutation have similar survival to non-carriers. Results from the Danish Breast Cancer Group [20] also confirmed that BRCA mutation was not associated with OS (adjusted HR: 1.98, 95% CI: 0.87–4.52, *p* = 0.10) while *BRCA1* breast cancer patients had shorter ten-year DFS than *BRCA2* BC patients. Based on these studies, BRCA mutation status was not regarded as an independent predictor for clinical outcome of breast cancer. Against this background, Wang et al. [21] reported that BRCA mutation carriers were more likely to be diagnosed with breast cancer with lymph node involvement (66.7% vs. 42.6%; *p* = 0.011) and had significantly worse breast-cancer-specific outcomes. The 5-year disease-free survival was 73.3% for BRCA mutation carriers and 91.1% for non-carriers (hazard ratio for recurrence or death 2.42, 95% CI 1.29–4.53; *p* = 0.013). After adjusting for clinical prognostic factors, BRCA mutation remained an independent poor prognostic factor for cancer recurrence or death (adjusted hazard ratio 3.04, 95% CI 1.40–6.58; *p* = 0.005). Non-BRCA gene mutation carriers did not exhibit any significant difference in cancer characteristics or outcomes compared to those without detected mutations. In the current study, we confirmed that 10-year OS rate did not show significant difference according to BRCA mutation status (non-mutated versus *BRCA1/2*-mutated tumors: 96.2% vs. 98.0%, *p* = 0.844), similar to previous studies. However, we proved that non-mutated tumors and *BRCA1/2*-mutated tumors showed significant difference in 10-year DFS rates (80.0% vs. 62.8%, *p* = 0.02, Figure 1). Specifically, *BRCA1* mutation was associated with decreased DFS (non-mutated versus *BRCA1*-mutated tumors: 78.6% vs. 62.1%, *p* = 0.031), while *BRCA2* mutation was not (*p* = 0.197). As a prognostic factor, clinical N stage remained a significant prognostic factor for DFS (HR: 2.078; 95% CI: 1.043–4.143; *p* = 0.038) on multivariate analysis (Table 4). BRCA mutation status remained a significant prognostic factor for contralateral breast RFS (HR: 4.155; 95% CI: 1.789–9.652; *p* = 0.001) on multivariate analysis (Table 4).

In terms of failure patterns, many studies have compared ipsilateral and/or contralateral tumor recurrence in *BRCA1* and *BRCA2* mutation carriers and patients with sporadic cancers. Studies have consistently identified an elevated risk of contralateral breast cancer in BRCA mutation carriers. Haffty et al. [15] have reported a recurrence rate of 42% in carriers versus 9% rate in non-carriers (*p* = 0.001) at 12 years. They suggest that the high risk of contralateral breast cancer in *BRCA1*/*BRCA2* mutation carriers must be taken into account when choosing treatment. In contrast, reports regarding whether the risk of ipsilateral recurrence is higher in women with a BRCA mutation have shown conflicting results. Kirova et al. [22] investigated whether mutation status can influence the rate of ipsilateral and contralateral breast cancers after breast-conserving treatment. They found no significant difference in rates of ipsilateral tumors among mutation carriers, non-carriers, and controls (*p* = 0.13). On multivariate analysis, age was the most significant predictor for ipsilateral recurrence (*p* < 0.001). The rate of contralateral cancer was significantly higher in familial cases: 40.7% (mutation carriers), 20% (non-carriers), and 11% (controls) (*p* < 0.001). After 13.4 years of follow-up, the rate of ipsilateral tumors was not higher in mutation carriers than that in non-carriers or controls. The present study also confirmed that *BRCA1/2*-mutated tumors showed a higher incidence of contralateral breast recurrence compared to non-mutated tumor (non-mutated versus *BRCA1/2*-mutated tumors: 4.9% vs. 26.0%, *p* < 0.001). Statistically-significant difference in 10-year contralateral breast RFS rates was found between non-detected and detected groups (95.0% vs. 73.7%, *p* < 0.001). However, no significant difference was found in the incidence of ipsilateral breast recurrence (4.5% vs. 6.0%, *p* = 0.649) or 10-year ipsilateral breast RFS rates (93.8% vs. 93.0%, *p* = 0.926). Additionally, there was no difference in the rate of regional recurrence (*p* = 0.991) or distant metastasis (*p* = 0.657) between the two groups.

Our current study has several limitations. First, it was a retrospective study and there might have been selection bias. Second, the sample size was too small to determine the statistical significance of associations between the two groups. Nonetheless, our current study demonstrated that patients having a BRCA mutation, especially *BRCA1* mutation, showed inferior DFS, although it did not reach statistical significance. BRCA mutation status was associated with higher risk of contralateral breast cancer. It was also a significant prognostic factor for contralateral breast RFS, although the rate of ipsilateral recurrence was not higher. Thus, clinicians should inform about the role of prophylactic contralateral mastectomy, prophylactic oophorectomy, tamoxifen administration, and/or active surveillance with close radiological surveillance to breast cancer patients with a BRCA mutation. Recently, PARP inhibitors have been investigated not only as chemo/radiotherapy sensitizers, but also as single agents to selectively kill cancers defective in DNA repair, specifically cancers with mutations in *BRCA1/2* genes.

## 5. Conclusions

Korean patients having a BRCA mutation, especially *BRCA1* mutation, showed inferior DFS, despite such interiority not reaching statistical significance. *BRCA1/2*-mutated tumors herald a higher risk of contralateral breast cancer. BRCA mutation status was a strong predictor of recurrence in contralateral breast. Strategies such as prophylactic treatment and active surveillance should be discussed with BRCA mutation-associated breast cancer patients. To determine the effect of *BRCA1* or *BRCA2* mutation on breast cancer survival, further study with larger scale is needed.

## Figures and Tables

**Figure 1 medicina-56-00514-f001:**
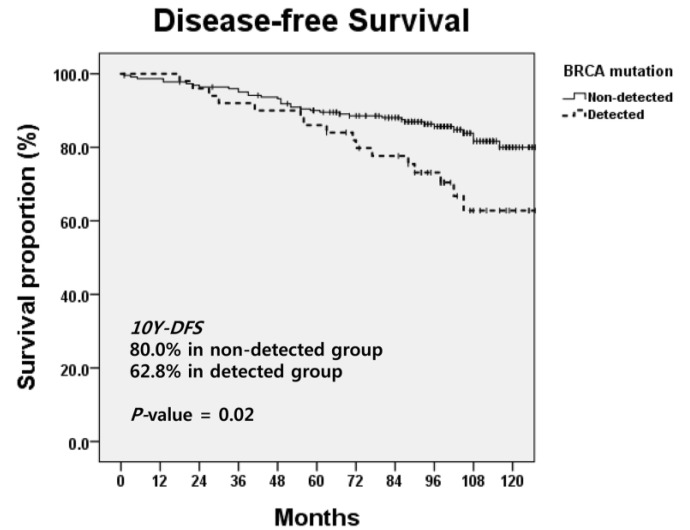
Disease-free survival curves based on the presence of a BRCA mutation.

**Table 1 medicina-56-00514-t001:** Patients, tumors, and treatment characteristics (*n* = 273).

Characteristics	Non-Mutated	*BRCA1/2*-Mutated Tumors (*n* = 50)	*p* Value
	Tumors (*n* = 223)		
Age years; median (range)		43 (21–70)	
Age group (years)			0.324
≤50	179 (80.3%)	37 (74.0%)	
>50	44 (19.7%)	13 (26.0%)	
Clinical T stage			0.554
Tis-2	204 (91.5%)	47 (94.0%)	
T3–4	19 (8.5%)	3 (6.0%)	
Clinical N stage			0.627
N0	139 (62.3%)	33 (66.0%)	
N1–3	84 (37.7%)	17 (34.0%)	
Hormonal receptor status			0.004
Negative	57 (25.6%)	23 (46.0%)	
Positive	166 (74.4%)	27 (54.0%)	
HER2/neu receptor status			0.008
Negative	187 (83.9%)	49 (98.0%)	
Positive	36 (16.1%)	1 (2.0%)	
Molecular subtypes			0.001
Luminal A	107 (48.0%)	17 (34.0%)	
Luminal B	59 (26.5%)	10 (20.0%)	
HER2-enriched	17 (7.6%)	1 (2.0%)	
Triple-negative	40 (17.9%)	22 (44.0%)	
Ki-67 status			0.013
Negative	113 (53.3%)	14 (32.6%)	
Positive	99 (46.7%)	29 (67.4%)	
Unknown	11	7	
Median number of sampled lymph nodes (LNs) (range)		8 (1–49)	
Pathology			0.625
DCIS	18 (8.0%)	5 (10.0%)	
IDC	177 (79.4%)	41 (82.0%)	
Others	28 (12.6%)	4 (8.0%)	
Pathologic T stage			0.755
(y)pTis	17 (7.6%)	5 (10.0%)	
(y)pT1	136 (61.0%)	28 (56.0%)	
(y)pT2	56 (25.1%)	15 (30.0%)	
(y)pT3	14 (6.3%)	2 (4.0%)	
Pathologic N stage			0.814
(y)pN0	143 (64.1%)	34 (68.0%)	
(y)pN1	57 (25.6%)	11 (22.0%)	
(y)pN2	14 (6.3%)	4 (8.0%)	
(y)pN3	9 (4.0%)	1 (2.0%)	
Histologic grade			0.012
grade 1–2	145 (65.0%)	23 (46.0%)	
grade 3	78 (35.0%)	27 (54.0%)	

DCIS = ductal carcinoma in situ; IDC = invasive ductal carcinoma.

**Table 2 medicina-56-00514-t002:** Survival and patterns of failure in BRCA1/2-mutated carriers versus non-carriers (*n* = 273).

Characteristics	Non-Mutated Tumors (*n* = 223)	BRCA1/2-Mutated Tumors (*n* = 50)	*p* Value
Survival outcomes			
10-year overall survival	96.20%	98.00%	0.844
10-year disease-free survival	80.00%	62.80%	0.02
Patterns of failure			
Local recurrence			
Ipsilateral	10 (4.5%)	3 (6.0%)	0.649
Contralateral	11 (4.9%)	13 (26.0%)	<0.001
Regional recurrence	9 (4.0%)	2 (4.0%)	0.991
Distant metastasis	14 (6.3%)	4 (8.0%)	0.657
Secondary cancer			
Ovarian cancer	5 (2.2%)	7 (14.0%)	0.001

**Table 3 medicina-56-00514-t003:** Univariate analysis for factors affecting survival outcomes (*n* = 273).

Characteristics	10 Year-DFS	*p* Value	Ipsilateral_RFS	*p* Value	Contralateral_RFS	*p* Value
Age group (years)		0.52		0.972		0.697
>50	75.30%		92.60%		89.90%	
≤50	77.10%		93.90%		91.30%	
Pathology		0.792		0.381		0.362
DCIS	72.90%		87.50%		87.50%	
IDC	76.20%		93.10%		92.20%	
Others	80.30%		100%		83.80%	
Clinical T stage		0.435		0.854		0.197
Tis-2	76.90%		93.60%		90.20%	
T3–4	75.90%		95.50%		100%	
Clinical N stage		<0.001		0.778		0.429
N0–1	79.70%		93.70%		90.90%	
N2–3	58.30%		93.60%		90.40%	
BRCA1/2 mutation status		0.02		0.926		<0.001
Non-mutated	80.00%		93.80%		95.00%	
Mutated	62.80%		93.00%		73.70%	
Hormonal receptor status		0.028		0.423		0.021
Negative	71.90%		92.90%		85.70%	
Positive	78.40%		93.80%		92.90%	
HER2/neu receptor status		0.36		0.586		0.929
Negative	75.60%		93.30%		90.80%	
Positive	84.90%		96.70%		91.10%	
Molecular subtypes						
Luminal A	80.60%	0.056	93.40%	0.504	93.80%	0.124
Luminal B	75.20%		95.40%		91.70%	
HER2-enriched	80.20%		100%		80.20%	
Triple-negative	69.60%		90.90%		87.10%	
Pathologic T stage		0.822		0.42		0.27
Tis-2	76.70%		93.40%		90.40%	
T3–4	78.70%		100%		100%	
Pathologic N stage		0.057		0.615		0.622
N0–1	78.60%		93.80%		91.00%	
N2–3	62.50%		92.20%		88.70%	
Ki-67 status		0.013		0.412		0.122
Negative	83.60%		93.10%		95.70%	
Positive	73.90%		93.90%		90.20%	
Histologic grade		0.182		0.553		0.035
grade 1–2	79.10%		93.70%		94.20%	
grade 3	73.10%		93.50%		86.00%	

DFS = disease-free survival; RFS = recurrence-free survival; DCIS = ductal carcinoma in situ; IDC = invasive ductal carcinoma.

**Table 4 medicina-56-00514-t004:** Multivariate analyses for factors affecting survival outcomes (*n* = 273).

Characteristics	Hazard Ratio	95% CI	*p* Value
DFS			
Clinical N stage (N2–3 vs. N0–1)	2.078	1.043–4.143	0.038
BRCA mutation status (mutated vs. non-mutated)	1.772	0.917–3.421	0.088
Hormonal receptor status (positive vs. negative)	0.779	0.395–1.537	0.471
Ki-67 status (positive vs. negative)	1.525	0.745–3.122	0.249
Contralateral RFS			
BRCA mutation status (mutated vs. non-mutated)	4.155	1.789–9.652	0.001
Hormonal receptor status (positive vs. negative)	0.699	0.262–1.868	0.476
Histologic grade (grade 3 vs. grade 1–2)	1.470	0.553–3.906	0.440

CI = confidence interval; DFS = disease-free survival; RFS = recurrence-free survival.

## Data Availability

The datasets used and/or analyzed during the current study are available from the corresponding author on reasonable request.

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
