# Peer review of "Germline BRCA Mutation and Clinical Outcomes in Breast Cancer Patients Focusing on Survival and Failure Patterns: A Long-Term Follow-Up Study of Koreans"

_medicina, 2020, doi:10.3390/medicina56100514_

Round 1

Reviewer 1 Report

Small numbers of patients but main usefulness of study is really looking at the Korean population - if there is any racial differences re: BRCA outcomes. Need larger sample size of patients to be able to truly apply the results Otherwise well written, good statistical analysis. Only issue (minor) I think it would be important to define the type of surgery (in particular) the BRCA carriers underwent as part of their breast cancer therapy. Obviously the type of surgery can influence rates of LR or contralateral recurrence.

Reviewer 2 Report

Comments to Kim et al.

This is an interesting and well-written paper, in line with journal’s aim and scope, where the main objective is to evaluate the effect of BRCA mutation on survival and recurrence rate in breast cancer (BC) patients who underwent genetic screening for BRCA1/2 main known mutations.

The research was executed according to currently accepted standards, approach to statistical analysis of data is appropriate and the manuscript provides data that are likely to be of interest to journal readers. Summarizing, the paper is interesting to read, however, some minor corrections should be applied and listed below:

  1. Please, use acronyms for sentences often used in the manuscript. The first time the acronym is used (in parenthesis), it should be fully written out.
  2. Please, provide affiliation number for all the authors.
  3. Please explain, in the introduction section, the characteristics of triple-negative BC tumors.
  4. As known, many factors could explain the discordant results among studies regarding the roles of BRCA1&2 mutations on BC survival and recurrence rate, including different population characteristics such as ER PgR and Her2/neu status, age, menopausal status, sample size, ethnicity and different genetic backgrounds. Thus, I strongly suggest to include some additional information (especially related to the ethnicity), for every study that the authors decided to include in introduction and discussion sections.
  5. No description of genetic procedures used to assay BRCA1&2 mutations was included in the materials and methods section. Please provide this relevant information.
  6. Please, provide the rs number of selected mutations assayed in this work.
  7. Considering that BC patients used in this work were treated used different treatment schemes (BCS, MRM, IM etc.), could these differences have a relevant role in BC survival and recurrence rates? Please deepen this topic and its possible relation to work results.
  8. Please, provide an additional table regarding treatment schemes used for all the BC patients included in this work, to facilitate the reader's comprehension.
  9. Which kind of RT schedule was used to treat BC patients?
  10. Please, mark in bold p-values related to interesting statistical significant correlations.
  11. Please add numbers in the reference section because is difficult to link the correct reference number (included in the text), and specific reference.
  12. Literature data report that in some geographic area, BC patients carriers of BRCA1&2 mutations going to anticipated screening respect to the non-carrier group. If possible, expand this topic including information especially related to the geographic area of BC patients used in this work.
